# Effects of Leisure Participation on Life Satisfaction in Older Korean Adults: A Panel Analysis

**DOI:** 10.3390/ijerph17124402

**Published:** 2020-06-19

**Authors:** Hyejin Yoon, Won Seok Lee, Kyoung-Bae Kim, Joonho Moon

**Affiliations:** 1Department of Global Tourism, Baewha Women’s University, 34 Pirundae-Ro 1-gil, Jongno-Gu, Seoul 03039, Korea; hyejin@baewha.ac.kr; 2Department of Tourism & Recreation, Kyonggi University, 24, Kyonggidae-Ro 9-gil, Seodaemun-Gu, Seoul 03746, Korea; lws798@kgu.ac.kr; 3Tourism Research Institute, Hanyang University, 413 College of Social Sciences, 222 Wangsimni-Ro, Seongdong-Gu, Seoul 04763, Korea; 4Department of Tourism Administration, Kangwon National University, 1 Kangwondaehak-Gil, Chuncheon, Kangwon-do 24341, Korea; joonhomoon0412@gmail.com

**Keywords:** older adults, leisure participation, life satisfaction, panel analysis

## Abstract

South Koreans’ life expectancy has dramatically increased over the last four decades. However, the life satisfaction index of older Korean adults has been in the bottom third globally. The large majority of older Koreans spend most of the day watching television at home. However, concrete evidence regarding the effects of leisure involvement on older adults’ quality of later life is scant. Only a few existing studies have examined the link via cross-sectional survey data. Thus, the purpose of this study was to investigate whether meaningful leisure participation outside the home in older age plays an essential role in improving life satisfaction. To achieve the research aim, nationally representative panel data from the Korea Employment Information Service were used for the data analysis. The results indicated that social and productive leisure participation in religious activity, social gatherings, and volunteering was significantly related to quality of life in older adults. Moreover, frequent participation in travel and cultural activities outside the home were positively related to life satisfaction. These findings suggest that participation in meaningful leisure activities is a critical factor contributing to subjective well-being and good mental health in older Korean adults and should be encouraged.

## 1. Introduction

South Korean life expectancy has increased remarkably over the past four decades, from 61.9 years in 1970 to 82.7 years in 2017, leading South Korea to be ranked 5th among the member countries of the Organization for Economic Cooperation and Development (OECD) [1]. Due to this increase in life expectancy, South Korea (hereafter Korea) became an “aging society”—defined as a country with 7% of its population aged 65 and older—in 2000, and an “aged society”—defined as a country in which older adults constitute 14% of the population—in 2017. The number of those in their 80s or older stood at 900,062, which was only 1.8% of the population, in 2010; this figure rose to 1.33 million (2.5%) in 2015 and to 1.92 million (3.7%) in 2020 [2]. In 2017, older Korean adults aged 60 and older felt more dissatisfied with their lives than those in other age groups [3]. In contrast, Statistics Canada (2017) found that Canadian older adults aged 65 and older are more satisfied with their lives than those in other age groups [4]. This research suggests that older adults are living more isolated lives, which could lead to loneliness [4]. Moreover, the Korean baby boomers, born between 1955 and 1963, started turning 65-years-old in 2020. Korea is expected to become a “super-aged society”—defined as a country where older adults constitute 20% of the population—by 2025, making it one of the fastest aging societies in the world [2].

The life expectancy of older Koreans has radically increased due to health-related advances. Nevertheless, statistics have shown that life satisfaction in this group is much lower than that in any other age group in Korea [2,3]. In prior studies, a U-shaped “age–happiness curve” with a high happiness index at the older and younger age points generally appears [5,6]; however, this pattern has not been observed for Korea. Life satisfaction is a subjective evaluation of one’s life as a whole based on the fit between personal goals and achievements [7,8]. It is generally considered an element and a critical indicator of quality of life or subjective well-being [8]. Life dissatisfaction is strongly related to mental health [7,8,9]. In this sense, some studies highlighted the association of loneliness with mental health problems, especially depression [9,10]. This mental disorder is significantly related to suicide globally [8]. In fact, the suicide rate among senior citizens in Korea was the highest among OECD members in 2015, with the happiness index for senior citizens being low enough to cause social problems [2].

The relevant literature indicates that Korean seniors’ leisure involvement may contribute to their life satisfaction, subjective well-being, and mental health [11,12]. Korean seniors, however, have not had the opportunity to form positive values regarding, and attitudes towards, leisure participation, so older individuals have little perception of how to spend their free time in their daily lives [13]. In their early years, they suffered from hunger and poverty during national crises, e.g., Japanese colonial rule and independence (1910–1945) and the Korean War (1950–1953), and endured postwar hardship. Negative attitudes toward leisure participation have persisted in Korea since the establishment of an industrial base after the Korean War in the 1960s and the industrial period in the 1970s and 1980s [13]. As a result, older adults had few opportunities to develop leisure skills, knowledge, or motivations during their childhood and adulthood. A lack of previous experience could negatively influence active leisure participation, ultimately limiting participation in leisure activities and reducing life satisfaction in older Koreans. As an example, the large majority of older Koreans are typically involved in indoor sedentary activity; over 99% of them engage in passive sedentary activity, e.g., watching T.V. [14].

The Korean government has neglected public leisure policy for older adults and their subjective well-being [15]. Korean society as a whole, including policymakers and researchers, still appears to lack a sense of urgency in coping with the demographic challenges of an aged society, particularly regarding subjective well-being in later life [13,15]. The older demographic has been devalued and marginalized by Korean society [13]. Thus, invisible components such as psychological and mental health in later life have been overlooked. To date, there is minimal research on issues related to older people in the Korean literature [15]. There is likely a significant connection between participation in leisure activities and older adults’ subjective well-being; however, a limited number of empirical studies have investigated the relationship [16,17]. According to the previous literature, not all leisure contributes to an individual’s life satisfaction and well-being, and the outcome might depend on the types of activities in which an individual engages [18]. Nevertheless, numerous studies have mainly focused on leisure-time physical activity and older adults’ physical and cognitive health [19,20,21], age-related diseases or mortality reduction [22,23].

Less is known about the degree to which leisure activity can influence aspects of subjective well-being, such as life satisfaction, in older adults [16]. Some research has shown that social and productive leisure activities (e.g., social activities, volunteering) are related to life satisfaction, but few studies focus on participation in this type of leisure activity and life satisfaction in older adults. Most Korean seniors are mainly engaged in sedentary leisure activities, which usually take place in solitude (e.g., television viewing, resting), and do not have a large leisure repertoire [13,14]. Only limited social and productive activities outside the home are observed among older Koreans, and these include social gatherings, volunteering, religious activity, and cultural activities [14]. In addition to investigating leisure activities, we review the relationship between travel frequency and life satisfaction among older adults. Despite the potential relevance of travel to life satisfaction, previous studies have focused mainly on older adults’ travel motivation or market segmentation from the perspective of the tourism industry [24]. Therefore, we focus on participation in social and productive activities, which contributes to improving life satisfaction, as an indicator of successful aging in older Korean adults. Furthermore, a limitation of the existing research is that it involves the analysis of longitudinal data. Thus, the causality of the observed relationships is still uncertain, and a careful review of the relationships is necessary. In this context, the aim of this study is to longitudinally examine how meaningful leisure activities are associated with life satisfaction in older Korean adults through a panel data analysis for the period 2006–2016.

## 2. Materials and Methods

### 2.1. Data Collection

Archival data were used for the data analysis. The data were derived from senior citizen research panel data collected by the Korea Employment Information Service. The Korea Employment Information Service has conducted basic research every two years since 2006. The study period of the research was between 2006 and 2016. We utilized five waves of panel data (2008, 2010, 2012, 2014, 2016), with identical questions regarding the leisure activities and life satisfaction asked in each wave. A total of 7490 participants were included in the analysis, yielding 37,450 observations. After data cleaning, 17,630 observations were ultimately used for the panel data analysis.

### 2.2. Explanation of the Variables

The dependent variable in this study is life satisfaction, which was measured through the participants’ self-evaluations of their degree of satisfaction with their lives on a scale from 1 to 100. Activity participation was used as a dummy variable (0 = no participation, 1 = participation) and included religious activity (REL), social gatherings (SOC), and volunteering (VOL). Frequency of annual travel (TRA) and cultural activities (e.g., movies, musicals, exhibitions, and sports) (CUL) were included in the study model as additional independent variables. Single-item measures were used: Do you participate in religious activity? Do you take part in any social gatherings? Do you participate in any volunteering activity? How frequently do you travel annually? How frequently do you participate in cultural activities annually? Furthermore, five control variables were included: birth year (AGE), sex (0 = male, 1 = female), monthly recreation expense (REC), monthly eating out expense (EOT), and individual annual total assets (AST).

### 2.3. Data Analysis

Descriptive analysis, correlation analysis, and multiple regression analysis were carried out to examine the data characteristics, the overall relation of the variables, and the likelihood of multicollinearity. To test the research hypotheses, ordinary least squares (OLS) regression analysis was initially performed. Then, two additional econometric analysis models were developed to check the consistency of the analysis results. First, a one-way fixed effects model, which incorporated multiple-year dummy variables into the regression model, was run. The aim of a one-way fixed effects model is to minimize bias from the year effect [25,26]. Then, a feasible generalized least squares (FGLS) model, which minimizes the bias caused by heteroskedasticity and autocorrelation, was run [27,28]. The regression equation was as follows:*LSit = β0 + β1RELit + β2SOCit + β3VOLit + β4RECit + β5EOTit + β6TRAit + β7CULit + β8AGEit + β9SEXit + β10ASTit + εit*
which includes life satisfaction (LS), religious activity (REL), social gatherings (SOC), volunteering (VOL), annual travel frequency (TRA), annual frequency of participation in cultural life (CUL), birth year (AGE), sex, recreation expense (REC), eating out expense (EOT), and individual total assets (AST).

## 3. Results

### 3.1. Descriptive Statistics

The results of the descriptive statistical analysis are presented in Table 1. The mean life satisfaction value is 62.628, and its standard deviation is 16.070. The mean REL, SOC, and VOL values are 0.163, 0.626, and 0.006, respectively. Table 1 also presents the descriptive statistics for TRA (Mean = 1.339, SD = 2.731), CUL (Mean = 0.754, SD = 2.200), REC (Mean = 5.065, SD = 9.287), and EOT (Mean = 9.935, SD = 10.527).

### 3.2. Correlation Matrix

Table 2 shows the results of the correlation matrix. Life satisfaction is positively correlated with REL (r = 0.032), SOC (r = 0.240), VOL (r = 0.030), REC (r = 0.202), EOT (r = 0.253), TRA (r = 0.155), CUL (r = 0.177), AGE (r = 0.240), and AST (r = 0.171). However, life satisfaction is negatively correlated with sex (r = −0.058). Regarding the correlation coefficients among the independent variables, none of the values are higher than 0.9, indicating that the likelihood of multicollinearity is fairly low [26].

### 3.3. Results of the Multiple Regression Analysis

Table 3 presents the results of the multiple regression analysis. Three types of econometric analyses were performed to check the reliability of the results. The three sets of results show consistency with respect to the significance and direction, and the F-statistics and Wald χ2 values of the models are significant. Specifically, REL (β = 1.675, *p* < 0.01), SOC (β = 4.567, *p* < 0.01), and VOL (β = 2.342, *p* < 0.05) are positively associated with life satisfaction. Additionally, life satisfaction is positively influenced by REC (β = 0.054, *p* < 0.01) and EOT (β = 0.128, *p* < 0.01). Moreover, life satisfaction is positively affected by TRA (β = 0.311, *p* < 0.01) and CUL (β = 0.413, *p* < 0.01). Regarding the control variables, AGE (β = 0.196, *p* < 0.01) and AST (β = 0.001, *p* < 0.01) are positively associated with life satisfaction. These results indicate that participation in meaningful leisure activities (i.e., volunteering, religious activity, social gatherings, travel, and cultural activities) had a higher life satisfaction in later life. Overall, all the proposed hypotheses are supported by the results of the analysis. Older adults with higher levels of expenditure on recreation and eating outside have considerably higher life satisfaction. Life satisfaction rises with the level of expenditure and total assets.

## 4. Discussion

This study provides empirical evidence regarding the relationship between participation in meaningful leisure activities and life satisfaction in older Korean adults by focusing on intra-individual change over time. Tests based on survey data from five waves of the Korean senior panel—a large, nationwide, randomly sampled panel study—showed that all social and productive leisure activities have a positive influence on life satisfaction. Meaningful leisure engagement among older adults was significantly associated with higher levels of life satisfaction. Specifically, participation in volunteering, religious activity, and social gatherings significantly predicts life satisfaction as an indicator of successful aging. More frequent involvement in leisure travel and cultural activities (e.g., movies, musicals, exhibitions, and sports) predicts higher life satisfaction in older Korean adults.

Older adults’ leisure participation in meaningful activities (i.e., religious activity, social gathering, and volunteering) was significantly related to life satisfaction. These findings are consistent with previous cross-sectional studies supporting the positive effect of religious engagement on individuals’ life satisfaction [29,30] and of social gatherings on older adults’ well-being [31]. This finding is also concordant with successful aging theory, which suggests that active engagement with life, including productive activities and interpersonal relations, can be a key factor for aging well [32,33,34]. Therefore, leisure service providers need to create community-based leisure programs such as volunteering opportunities, hobby/arts/sports-related clubs or gatherings. In particular, Korean government policy needs to advocate the commitment of volunteering as a method of improving life satisfaction and civic engagement for older adults. Promoting volunteering can be a great public health promotion intervention for older adults to increase physical and mental health outcomes [35]. However, it is unclear which type of volunteering activity is related to the greatest life satisfaction and well-being in later life. Future research should explore the types of volunteering activities we could not include in our study, such as formal and informal volunteering.

Additionally, our study is an initial investigation of the relationship between cultural activities (e.g., movies, musicals, exhibitions, and sports) and life satisfaction in the older Korean population, although many studies have shown that leisure engagement contributes to health and life satisfaction [11,12,29,30,31,32,33,34]. The frequency of cultural activities was found to play an important role in the life satisfaction of older adults. Thus, the finding from this study suggests that cultural activity interventions and community programs designed to target older adults have the potential to improve life satisfaction and subjective well-being.

Frequent participation in leisure travel was a significant predictor of life satisfaction. Encouraging travel experiences could be a promising approach to promoting older Koreans’ perceived quality of life. However, little is known about the association between older adults’ travel experience and perceived quality of life, including life satisfaction [15]. This is because most previous studies related to senior travel have employed an etic approach, focusing heavily on the promising senior market as a niche for those with disposable income, older individuals’ travel motivation and the constraints for tourism suppliers and marketers, e.g., [36,37,38,39,40]. Thus, our empirical evidence calls for future studies to confirm the link between older adults’ travel experiences and life satisfaction, and to elaborate on this association in the context of individual, cultural, and generational differences. In this context, researchers need to shift the focus of senior travel research from the marketing perspective to older individuals’ subjective well-being (i.e., an emic perspective); for instance, future researchers should consider how older adults perceive their later lives through travel experience. Which types of travel have meaning for relatively healthy older adults or older adults who are from different social classes, cultures, or races; have different resources; or have specific physical limitations or diseases? How can marginalized older adults imagine and be involved in this leisure activity? Regarding this line of research, leisure providers should also consider older adults’ subjective well-being by inviting older adults to participate in their tourism services before the services are produced, provided, and marketed. This approach will enable leisure providers to design more appropriate products and services for each group of older adults (e.g., those from higher social classes and lower social classes and those with specific physical, psychological, and cognitive limitations or diseases). Furthermore, in terms of the association between older adults’ socioeconomic and health status and travel participation, more marginalized people (e.g., lower-class individuals and those with physical limitations) may not be able to easily participate in leisure travel. Thus, leisure providers and policymakers need to consider this issue and provide tourism opportunities as a welfare service for them; these new perspectives and efforts could contribute to marginalized older adults’ aging well.

Nevertheless, watching television is still the most popular type of leisure in later life; older adults watch more television than any other age group worldwide [41]. Given the notable role of television viewing in older adults’ lives, few studies have focused only on the impact of television viewing on older adults’ health outcomes, such as obesity and the risk for a specific disease (e.g., cardiovascular disease) or mortality, e.g., [42,43,44]. Therefore, future research should identify the influence of this sedentary activity in which older people commonly engage on diverse aspects of quality of life (e.g., physical, psychological, and social well-being).

Despite the strength of the representative sample and the longitudinal design, this study has several limitations. First, the main limitation of this study is the limited number of leisure activities in the analysis and the rough measurement of involvement in activities. Due to the limited survey questionnaire and collected data, independent variables that may affect the satisfaction of older adults may have been missed. Thus, further research should examine panel data for the diverse types of activities that we could not include in this study, such as socializing with the family. Additionally, these quantitative data could not provide an in-depth investigation or explanation of older adults’ perceptions and inner feelings about their quality of life and well-being. Further qualitative research is needed to thoroughly investigate older adults’ leisure behavior and its impact on their later lives and successful aging. Moreover, this study focused on the relationship between leisure activities and life satisfaction. It is possible that there are other important factors that influence older adults’ leisure involvement and subjective well-being, such as income level, personal health status, and leisure attitude. Therefore, future studies need to extend this analysis by considering these influential factors to better understand and promote older adults’ quality of life and determine how older individuals’ characteristics (e.g., sociodemographics, health status, leisure attitude, and health promotion behavior) are related to their leisure involvement in various age-appropriate activities and life satisfaction.

## 5. Conclusions

Despite the importance of meaningful leisure engagement and increased free time in later life, many older Koreans do not participate in leisure activities outside their homes. Through three regression analyses for the purpose of a robustness check, this study highlights the benefits of leisure participation through social and productive activities (i.e., volunteering, religious activity, travel, social gatherings and cultural activities) and its association with an indicator of subjective well-being—life satisfaction. Clearly, life satisfaction is critical to achieving positive mental health and successful aging, indicating that mental health is as important as physical health in the older population. This evidence suggests that public policy for older people should create and deliver diverse, meaningful programs and provide leisure education to foster the development of leisure values, knowledge, skills, and attitudes and thereby increase the quality of later life.

## Figures and Tables

**Table 1 ijerph-17-04402-t001:** Descriptive statistics.

Variables	Mean	Standard Deviation
Life Satisfaction	62.628	16.070
REL	0.163	0.369
SOC	0.626	0.484
VOL	0.006	0.079
TRA	1.339	2.731
CUL	0.754	2.200
AGE	1947.909	10.335
SEX	0.574	0.494
REC (Unit: ten thousand KRW)	5.065	9.287
EOT (Unit: ten thousand KRW)	9.935	10.527
AST (Unit: ten thousand KRW)	21,227.480	31,863.760

Note: Religious activity (REL) (0 = No, 1 = Yes), Social gatherings (SOC) (0 = No, 1 = Yes), Volunteering (VOL) (0 = No, 1 = Yes), Annual travel frequency (TRA), Annual cultural life participation frequency (CUL), Birth year (AGE), Sex (0 = Male, 1 = Female), Recreation expense (REC) (Unit: ten thousand KRW), Eating out expense (EOT) (Unit: ten thousand KRW), and Individual total assets (AST) (Unit: ten thousand KRW).

**Table 2 ijerph-17-04402-t002:** Correlation matrix.

Variables	2	3	4	5	6	7	8	9	10	11
1. LS	**0.032**	**0.240**	**0.030**	**0.202**	**0.253**	**0.155**	**0.177**	**0.240**	**−0.058**	**0.171**
2. REL	1	**−0.120**	**0.061**	**0.045**	**0.025**	**0.014**	**0.067**	0.006	**0.118**	**−0.017**
3. SOC		1	0.009	**0.183**	**0.220**	**0.145**	**0.129**	**0.264**	**−0.078**	**0.111**
4. VOL			1	**0.110**	**0.054**	**0.029**	**0.068**	**0.036**	**0.031**	**0.012**
5. REC				1	**0.476**	**0.258**	**0.344**	**0.259**	**−0.027**	**0.284**
6. EOT					1	**0.233**	**0.307**	**0.409**	**−0.058**	**0.256**
7. TRA						1	**0.263**	**0.186**	**−0.024**	**0.100**
8. CUL							1	**0.265**	**0.029**	**0.158**
9. AGE								1	**−0.037**	**0.068**
10. SEX									1	**−0.163**
11. AST										1

Note: Bold indicates *p* < 0.05; Life satisfaction (LS), Religious activity (REL) (0 = No, 1 = Yes), Social gatherings (SOC) (0 = No, 1 = Yes), Volunteering (VOL) (0 = No, 1 = Yes), Recreation expense (REC) (Unit: ten thousand KRW), Eating out expense (EOT), Annual travel frequency (TRA), Annual cultural life participation frequency (CUL), Birth year (AGE), Sex (0 = Male, 1 = Female), and Individual total assets (AST) (Unit: ten thousand KRW).

**Table 3 ijerph-17-04402-t003:** Regression analysis results.

Variables	Ordinary Least Squares (OLS) β (t-Value)	Fixed Effects β (t-Value)	Feasible Generalized Least Squares (FGLS) β (Wald)
Intercept	−326.132 (14.62) **	−326.131 (−13.95)	−326.132 (14.62) **
REL	1.675 (6.07) **	1.675 (5.84) **	1.675 (6.07) **
SOC	4.567 (20.10) **	4.567 (19.51) **	4.567 (20.10) **
VOL	2.342 (2.13) *	2.342 (2.78) **	2.342 (2.13) *
REC	0.054 (4.29) **	0.054 (4.99) **	0.054 (4.29) **
EOT	0.128 (10.71) **	0.128 (10.58) **	0.128 (10.71) **
TRA	0.311 (8.15) **	0.312 (7.78) **	0.312 (8.16) **
CUL	0.413 (8.79) **	0.413 (11.57) **	0.413 (8.79) **
AGE	0.196 (17.10) **	0.196 (16.31) **	0.196 (17.10) **
SEX	−0.397 (−1.77)	−0.397 (−1.72)	−0.397 (−1.77)
AST	0.001 (11.92) **	0.001 (10.74) **	0.001 (11.92) **
F-value	290.43 **	231.77 **	
Wald χ^2^	-	-	2905.95 **
R-square	0.1279	0.1279	

Note: Dependent variable: Life satisfaction (LS); Independent variables: Religious activity (REL) (0 = No, 1 = Yes), Social gatherings (SOC) (0 = No, 1 = Yes), Volunteering (VOL) (0 = No, 1 = Yes), Annual travel frequency (TRA), and Annual cultural life participation frequency (CUL); Control variables: Birth year (AGE), Sex (0 = Male, 1 = Female), Eating out expense (EOT) (Unit: ten thousand KRW), Recreation expense (REC) (Unit: ten thousand KRW), and Individual total assets (AST) (Unit: ten thousand KRW). * *p* < 0.05, ** *p* < 0.01.

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
