# Peer review of "Effects of Leisure Participation on Life Satisfaction in Older Korean Adults: A Panel Analysis"

_ijerph, 2020, doi:10.3390/ijerph17124402_

Round 1

Reviewer 1 Report

  1. The new version have added some literature and have improved the meaning. However, the contribution (academic and practice) was still week. The analysis is still so simple. That why authors mentioned that "the aim of this study is to longitudinally examine how meaningful leisure activities are associated with life satisfaction in older Korean adults through a panel data analysis for the period 2006-2016." (Line 102-104). I can not find the answer "how".
  2. I hope authors can describe more about what contribute to practice.

Author Response

We would like to thank you for this opportunity to revise and resubmit our study. Please see our detailed responses below. 

1. The new version have added some literature and have improved the meaning. However, the contribution (academic and practice) was still week. I hope authors can describe more about what contribute to practice.

Response: Thank you for mentioning this point. Reflecting the reviewer’s opinion, we have added the contributions.

p.6 (2nd paragraph)

Older adults’ leisure participation in meaningful activities (i.e., religious activity, social gathering, and volunteering) was significantly related to life satisfaction. These findings are consistent with previous cross-sectional studies supporting the positive effect of religious engagement on individuals’ life satisfaction [29, 30] and of social gatherings on older adults’ well-being [31]. This finding is also concordant with successful aging theory, which suggests that active engagement with life, including productive activities and interpersonal relations, can be a key factor for aging well [32-34]. Therefore, leisure service providers need to create community-based leisure programs such as volunteering opportunities, hobby/arts/sports-related clubs or gathering. In particular, Korean government policy need to advocate the commitment of volunteering as a method of improving life satisfaction and civic engagement for older adults. Promoting volunteering can be a great public health promotion intervention for older adults to increases physical and mental health outcomes [35]. However, it is unclear which type of volunteering activity is related to the greatest life satisfaction and well-being in later life. Future research should explore types of volunteering activities we could not include in our study such as formal and informal volunteering.

p.6 (3rd paragraph)

Additionally, our study is an initial investigation of the relationship between cultural activities (e.g., movies, musicals, exhibitions, and sports) and life satisfaction in older population, although many studies have showed that leisure engagement contributes to health and life satisfaction [11, 12, 29-34]. The frequency of cultural activities was found to play an important role in life satisfaction of older adults. Thus, the finding from this study suggests that cultural activity interventions and community programs designed to target older adults have the potential to improve life satisfaction and subjective well-being.

2. The analysis is still so simple. That why authors mentioned that "the aim of this study is to longitudinally examine how meaningful leisure activities are associated with life satisfaction in older Korean adults through a panel data analysis for the period 2006-2016." (Line 102-104). I can not find the answer "how".

Response: First of all, we apologize for not giving you a satisfactory answer. However, rather than simply identifying the relationship between variables by using fragmentary data alone, this study is different from existing studies in that it attempted to more dynamically understand the relationship between variables through data that has changed over time. Please consider this generously.

Reviewer 2 Report

Although the paper has been improved I still have questions about the longitudinal aspect of the data and the analysis.

In my previous review  I asked:

Are there 5 measurements (2008, 2010, 2012, 2014, 2016) for each individual? If so, what is presented in table 1 and table 2?

And if so, wouldn't it be interesting to show how the variables change over time?

(And if not, what makes it longitudinal data instead of 5 times cross sectional)

How to interpret the std in table 1. Variability within the group or variability over time? Or a combination of both?

Similar questions for table 2, how is dealt with the five years in calculating the correlations?

By giving more insight in the development over time for at least Life Satifaction the discussion and the conclusions could be richer. Does for instance particpating in CUL help to maintain LS or help to improve LS or help to slow down a descrease in LS etc.

Is it necessary to present the data from three different regression analysis techniques? And if so, please reflect on it in the discussion section.

I saw that at least one of the other reviewers had similar questions about the longitudinal data:

  • Authors used a ten-year longitudinal research why? What benefit for use longitudinal research approach is? What research gap make authors use this method?
  • There have five-year panel data, why authors do not to test the difference of the five-year data? Check the trend of the five-year?

Since there are (in my opinion) no attempts to adress these topics of reviewers in the revised paper I had expected a point by point reply to answer the  questions of the reviewers.

Author Response

We would like to thank you for this opportunity to revise and resubmit our study.  We know how long it takes to review and critique a paper, particularly when providing comments as thoughtful as those we received. Please see our detailed responses below.

1. Although the paper has been improved I still have questions about the longitudinal aspect of the data and the analysis. In my previous review I asked: Are there 5 measurements (2008, 2010, 2012, 2014, 2016) for each individual? If so, what is presented in table 1 and table 2? And if so, wouldn't it be interesting to show how the variables change over time?

Response: Thank you for the reviewer’s comment. We fully understand what the reviewer mentioned. However, in panel/longitudinal data studies, it is common to look at the mean and standard deviation of variables as a whole. This is not only there is a large amount of panel/longitudinal data, but also it is not significant to repeatedly show the descriptive results that provide simple data information. Therefore, we also provide an average of 5 years of descriptive statistics.

2. And if not, what makes it longitudinal data instead of 5 times cross sectional) How to interpret the std in table 1. Variability within the group or variability over time? Or a combination of both?

Response: Thank you for your valuable comment. The standard deviation in this study means ‘combination of body’. This indicates how much the characteristics of the variable change over time that the panel/longitudinal data has.

3. Similar questions for table 2, how is dealt with the five years in calculating the correlations?

Response: Thank you for valuable comment. Similar to the above answer, the correlation was determined by the mean value in the analysis of correlation in terms of (1) the vast amount of data and (2) the repeated presentations of simple correlation.

4. By giving more insight in the development over time for at least Life Satifaction the discussion and the conclusions could be richer. Does for instance particpating in CUL help to maintain LS or help to improve LS or help to slow down a descrease in LS etc.

Response: Thank you for mentioning this point. Reflecting the reviewer’s opinion, we have added this point in the discussion.  

p.6 (3rd paragraph)

Additionally, our study is an initial investigation of the relationship between cultural activities (e.g., movies, musicals, exhibitions, and sports) and life satisfaction in older population, although many studies have showed that leisure engagement contributes to health and life satisfaction [11, 12, 29-34]. The frequency of cultural activities was found to play an important role in life satisfaction of older adults. Thus, the finding from this study suggests that cultural activity interventions and community programs designed to target older adults have the potential to improve life satisfaction and subjective well-being.

Also, we have added the contributions.  

p.6 (2nd paragraph)

Older adults’ leisure participation in meaningful activities (i.e., religious activity, social gathering, and volunteering) was significantly related to life satisfaction. These findings are consistent with previous cross-sectional studies supporting the positive effect of religious engagement on individuals’ life satisfaction [29, 30] and of social gatherings on older adults’ well-being [31]. This finding is also concordant with successful aging theory, which suggests that active engagement with life, including productive activities and interpersonal relations, can be a key factor for aging well [32-34]. Therefore, leisure service providers need to create community-based leisure programs such as volunteering opportunities, hobby/arts/sports-related clubs or gathering. In particular, Korean government policy need to advocate the commitment of volunteering as a method of improving life satisfaction and civic engagement for older adults. Promoting volunteering can be a great public health promotion intervention for older adults to increases physical and mental health outcomes [35]. However, it is unclear which type of volunteering activity is related to the greatest life satisfaction and well-being in later life. Future research should explore types of volunteering activities we could not include in our study such as formal and informal volunteering.

5. Is it necessary to present the data from three different regression analysis techniques? And if so, please reflect on it in the discussion section.

Response: Three different regression analyses were conducted to determine whether the bias in estimating the coefficients in this study has significantly changed the results of the study, and we wanted to confirm the above issues. This was briefly mentioned in the discussion and discussion sections.

6. I saw that at least one of the other reviewers had similar questions about the longitudinal data: Authors used a ten-year longitudinal research why? What benefit for use longitudinal research approach is? What research gap make authors use this method?

Response: According to various literature, the representative advantages of longitudinal data analysis is that it contains much more detailed information than information that can be collected through a one-time survey. Also, the use of time-changing data for analysis enables researchers to identify more dynamic relationships with research subjects, which could provide more insight into causal processes (Ap, 1990; Getz, 1994; Carmichael, Peppard, & Boudreau, 1996). So far, few studies that demonstrate leisure roles have employed the longitudinal data analysis. This is therefore assumed to be the methodological contribution of this study. Please consider this generously.

7. There have five-year panel data, why authors do not to test the difference of the five-year data? Check the trend of the five-year?

Response: The fixed-effect regression analysis indirectly identify the question that the reviewer has, but the results of the fixed effect is not statistically significant, representing that there seems to be no specificity between the data for five years in terms of the trend in the study.

Reviewer 3 Report

First of all, I would like to congratulated the authors for the changes and improvement of the work. Nevertheless, I have some recommendations. 

  1. Introduction: the introduction has been improved giving it more flow and structured.
    • Line 55 after the references, the authors could included "In this sense, some studies have highlighted ". 
    • Line 61, the references should placed after the entire sentence. 
  2. The results could be further explained, but the issue is with EOT (line 152), that is later explained in line 182. Please change this

Author Response

1. Introduction: the introduction has been improved giving it more flow and structured. Line 55 after the references, the authors could included "In this sense, some studies have highlighted ".

Response: We appreciate the comment. Based on the reviewer’s advice, we have improved the flow.

2. Line 61, the references should placed after the entire sentence.

Response: Thank you for mentioning this point. We have changed that. 

3. The results could be further explained, but the issue is with EOT (line 152), that is later explained in line 182. Please change this. 

Response: Thank you for mentioning this point. Based on the comment, we have added the result before the Table 3.

This manuscript is a resubmission of an earlier submission. The following is a list of the peer review reports and author responses from that submission.

Round 1

Reviewer 1 Report

ID: IJERPH-786010

Title: Effects of leisure participation on life satisfaction in older Korean adults: A panel analysis

Using data from a Korean senior citizen panel study, authors wanted to test whether meaningful leisure participation in older age plays an essential role in improving life satisfaction. The results revealed that social and productive leisure participation in religious activity, social gathering, and volunteering was significantly related to quality of life in older adults. Moreover, frequent participation in travel and cultural activities outside the home was positively related to their life satisfaction. The major shortcoming of this manuscript is nothing new, the analysis is so little and simple. Thus, the contribution (academic and practice) was very week. Following suggestion can be for referring to improve:

  1. First the abstract need to rewrite to become informative, and clearly point out what contribution is and what can be used in practice
  2. The relationship between leisure participation and well-being in older adults have long been studied. Authors should make a detailed literature review to show: how this problem was addressed in previous and to point out: What pros and cons of those methods? What problems still need to be improved? Thus, the introduction needs to present the research gap, motivation and how the authors want to contribute to it. A summary table comparing the contributions of literature could support the explanation.
  3. Large of productive or meaningful leisure activities, why authors only selected the volunteering, religious engagement, cultural activities, travel, and social gathering activities? Socializing with the family was very important, why ignore it?
  4. Authors used a ten-year longitudinal research why? What benefit for use longitudinal research approach is? What research gap make authors use this method?
  5. There have five-year panel data, why authors do not to test the difference of the five-year data? Check the trend of the five-year?
  6. Why do not analyze the relationship of degree of satisfaction with participation one or two or three…leisure activities?
  7. The contribution and discussion must connect with what the status quo description in introduction.
  8. This manuscript is nothing new, the analysis is so little and simple. Thus, the contribution (academic and practice) was very week. So I regard to recommend reject.

Reviewer 2 Report

I read the paper with pleasure. The topic is interesting and relevant.

I have questions rather than comments:

Mainly about the longitudinal aspect of the data and the analysis of the data:

Are there 5 measurements (2008, 2010, 2012, 2014, 2016) for each individual? If so, what is presented in table 1 and table 2?

And if so, wouldn't it be interesting to show how the variables change over time?

(And if not, what makes it longitudinal data instead of 5 times cross sectional)

How to interpret the std in table 1. Variability within the group or variability over time? Or a combination of both?

Similar questions for table 2, how is dealt with the five years in calculating the correlations?

By giving more insight in the development over time for at least Life Satifaction the discussion and the conclusions could be richer. Does for instance particpating in CUL help to maintain LS or help to improve LS or help to slow down a descrease in LS etc.

Is it necessary to present the data from three different regression analysis techniques? And if so, please reflect on it in the discussion section.

Check the sentence in the conclusions, it seems there are verbs missing:

Our findings suggested that the benefits of leisure participation through social and productive activities (i.e.volunteering, religious activity, travel, social gathering and cultural activity) and its association with indicators of successful aging—life satisfaction. 

'Eating Out Expense' is not consistently abbreviated. EOU and EOT are used.

Reviewer 3 Report

The authors have presented an interesting manuscript, although some changes must me applied to the manuscript.

I have uploaded a file with comments an suggestions to the authors.
